# SARS-CoV-2 Specific Immune Response and Inflammatory Profile in Advanced HIV-Infected Persons during a COVID-19 Outbreak

**DOI:** 10.3390/v14071575

**Published:** 2022-07-20

**Authors:** Alessandra Vergori, Antonio Boschini, Stefania Notari, Patrizia Lorenzini, Concetta Castilletti, Francesca Colavita, Giulia Matusali, Eleonora Tartaglia, Roberta Gagliardini, Andrea Boschi, Eleonora Cimini, Markus Maeurer, Pierluca Piselli, Leila Angeli, Andrea Antinori, Chiara Agrati, Enrico Girardi

**Affiliations:** 1HIV/AIDS Unit, National Institute for Infectious Diseases Lazzaro Spallanzani IRCCS, 00149 Rome, Italy; patrizia.lorenzini@inmi.it (P.L.); roberta.gagliardini@inmi.it (R.G.); andrea.antinori@inmi.it (A.A.); 2Medical Center, San Patrignano Community, 47853 Rimini, Italy; aboschini@sanpatrignano.org (A.B.); langeli@sanpatrignano.org (L.A.); 3Cellular Immunology and Pharmacology Laboratory, National Institute for Infectious Diseases Lazzaro Spallanzani IRCCS, 00149 Rome, Italy; stefania.notari@inmi.it (S.N.); eleonora.tartaglia@inmi.it (E.T.); eleonora.cimini@inmi.it (E.C.); chiara.agrati@inmi.it (C.A.); 4Laboratory of Virology, National Institute for Infectious Diseases, Lazzaro Spallanzani IRCCS, 00149 Rome, Italy; concetta.castilletti@inmi.it (C.C.); francesca.colavita@inmi.it (F.C.); giulia.matusali@inmi.it (G.M.); 5Infectious Disease Unit, Hospital of Rimini “Gli Infermi”, 47923 Rimini, Italy; boschia@libero.it; 6Immunotherapy Programme, Champalimaud Centre for the Unknown, 1400-038 Lisbon, Portugal; maeurer@mail.uni-mainz.de; 7I Medical Clinic, University of Mainz, 55122 Mainz, Germany; 8Clinical Epidemiology Unit, Department of Epidemiology and Preclinical Research, National Institute for Infectious Diseases Lazzaro Spallanzani IRCCS, 00149 Rome, Italy; pierluca.piselli@inmi.it; 9Health Direction, National Institute for Infectious Diseases Lazzaro Spallanzani IRCCS, 00149 Rome, Italy; 10Scientific Direction, National Institute for Infectious Diseases Lazzaro Spallanzani IRCCS, 00149 Rome, Italy; enrico.girardi@inmi.it

**Keywords:** advanced HIV infection, COVID-19, immunity, inflammation, outbreak, SARS-CoV-2

## Abstract

The main aim of this study was to describe the clinical and immunological outcomes, as well as the inflammatory profile, of patients with advanced HIV in an assisted-living facility in which an outbreak of SARS-CoV-2 occurred. SARS-CoV-2 humoral and specific T-cell response were analyzed in patients with HIV infection and COVID-19; as a secondary objective of the analysis, levels of the inflammatory markers (IL-1β, IL-6, IL-8, and TNFα) were tested in the HIV/COVID-19 group, in HIV-positive patients without COVID-19, and in HIV-negative patients with mild/moderate COVID-19. Antibody kinetics and ability to neutralize SARS-CoV-2 were evaluated by ELISA assay, as well as the inflammatory cytokines; SARS-CoV-2 specific T-cell response was quantified by ELISpot assay. Mann–Whitney or Kruskal–Wallis tests were used for comparisons. Thirty patients were included with the following demographics: age, 57 years old (IQR, 53–62); 76% male; median HIV duration of infection, 18 years (15–29); nadir of CD4, 57/mmc (23–100) current CD4 count, 348/mmc (186–565). Furthermore, 83% had at least one comorbidity. The severity of COVID-19 was mild/moderate, and the overall mortality rate was 10% (3/30). Additionally, 90% of patients showed positive antibody titers and neutralizing activity, with a 100% positive SARS-CoV-2 specific T-cell response over time, suggesting the ability to induce an effective specific immunity. Significantly higher levels of IL-6, IL-8, and TNF-α in COVID-19 without HIV vs. HIV/COVID-19 patients (*p* < 0.05) were observed. HIV infection did not seem to negatively impact COVID-19-related inflammatory state and immunity. Further data are mandatory to evaluate the persistence of these immunity and its ability to expand after exposure and/or vaccination.

## 1. Introduction

To date, the SARS-CoV-2 infection has not spared people living with HIV (PLWH). Case series published to date have not shown clear evidence for a higher infection rate of SARS-CoV-2 or a different clinical presentation in people with and without HIV [1,2,3,4,5,6,7,8,9,10,11,12]. The incidence of COVID-19 among PLWH seems to be more influenced by the differential impact of comorbidities and social determinants of health rather than HIV itself [13]. These case series of PLWH, however, were largely underpowered and often reported a younger age than in HIV-negative hospitalized COVID-19 patients [1,2,3,4,5,6,7,8,9,10,11,12]. Moreover, they did not focus on different immunodeficiency stages; indeed, the impact of coinfection with SARS-CoV-2 in PLWH remains incompletely understood. The picture is different if we look at the mortality risk; in fact, several studies have reported a higher mortality rate due to COVID-19 in a large cohort of PLWH in South Africa [14], the UK [15,16], and the US [17]. The main objective of this study was to describe the clinical outcomes and the inflammatory profile of patients with advanced HIV and SARS-CoV-2 coinfection in an assisted-living facility in which an outbreak occurred and to compare them with HIV-negative COVID-19 patients. The secondary objective was to verify the impact of advanced HIV infection on the plasmatic inflammatory profile during SARS-CoV-2.

## 2. Materials and Methods

### 2.1. Study Population

For this study, we identified three groups of patients. In group 1, we included advanced HIV-positive, >18 years old, virologically suppressed patients living in the rehabilitation facility of San Patrignano (Rimini, Italy), with SARS-CoV-2 infection diagnosed by means of a positive RT-PCR on nasopharyngeal swab (at least once), where an outbreak occurred between September and November 2020. The San Patrignano Center is a rehabilitation facility in which, at admission, people sign a release on the use of their data and samples, in anonymous and aggregate form, for research or statistical processing purposes. For this specific study, ethical review and approval were not required for the study on human participants in accordance with the local legislation and institutional requirements. The patients/participants or their caregivers, as appropriate, provided additional written informed consent to participate in this study.

Group 2 included advanced HIV-positive patients, >18 years old, followed up at the HIV/AIDS Unit of National Institute for Infectious Diseases L. Spallanzani in Rome, Italy, who were virologically suppressed, and for whom a blood sample was available and stored at our biological bank for other research purposes (consecutively enrolled in another observational study approved by our local ethical committee (RetroSNC, number 78/2016 approved on 14 June 2016)).

For both groups, a manual registration review of the medical record data was performed by one clinician/reviewer who recorded demographic data, HIV history information, presenting symptoms and signs, and general laboratory and HIV parameters at the time of evaluation and during the COVID-19 presentation for group 2.

Group 3 was composed of patients, >18 years old, who were consecutively admitted to the National Institute for Infectious Diseases L. Spallanzani in Rome, Italy, with SARS-CoV-2 infection diagnosed by means of RT-PCR positive on nasopharyngeal swabs (at least once) and/or serology from 1 March up to 31 July 2020 and for whom blood samples were available and stored at our biological bank. Data were collected for the ReCOVeRI Study, a registry on COVID-19 for clinical research of the National Institute for Infectious Diseases L. Spallanzani, approved by the Ethical Committee of the National Institute for Infectious Diseases L. Spallanzani IRCCS (number 164, 26 June 2020); the study was conducted in accordance with the Helsinki declaration. The management of the registry was adapted according to the standards of EUnetHTA reported in the Registry Evaluation and Quality Standards Tool (EUnetHTA, 2019). Both the laboratories and the data collection methods are the same, as previously reported in another study performed at our institute [18]. All methods were performed in accordance with the relevant guidelines and regulations. All patients or their caregivers, as appropriate, gave informed consent for collecting personal data for research purposes.

#### Definitions

Using the World Health Organization (WHO) HIV/AIDS revised clinical staging [19], all HIV patients with COVID-19 (*n* = 30) included in the analysis met the criteria for advanced disease, 15 of which were defined by immunological criteria (CD4 < 350/mmc) and 15 of which were defined by clinical criteria (clinical stage 3 or 4).

For groups 1 and 2, we recorded HIV history including the duration of infection, mode of HIV acquisition, AIDS diagnosis, self-reported or clinician-reported nadir CD4^+^ T-cell count, the most recent CD4^+^ T-cell count, and the last plasma HIV RNA level, as well as the most recent documented ART regimen, comorbidities, and clinical notes. Demographic, epidemiological, and clinical data, comorbidities, blood exams, therapeutic data about steroids and remdesivir use, presence of key symptoms of COVID-19 at the initial assessment, and oxygen saturation by measurement of pulse oximetry were collected for groups 1 and 3. The duration of viral shedding defined as the time from first positive to first negative swab was evaluated. Laboratory values of inflammatory markers, and the clinical outcomes were also recorded. When looking at and reviewing the database of patients in group 1, we noticed that all of them had a mild–moderate occurrence of COVID-19 (according to NIH criteria) [20]; for this reason, we decided to include only patients with mild/moderate pneumonia among patients of group 3 with COVID-19 and without HIV infection, in order to compare two homogeneous groups (group 1 vs. group 3) in terms of clinical and immunological markers.

### 2.2. Statistical Analyses

We summarized the general characteristics of three groups using median and interquartile range (IQR) for continuous parameters and absolute and percentage frequency for categorical variables. To compare groups, groups 1 vs. 2 and groups 2 vs. 3, we used χ^2^ or Fisher exact tests for categorical variables and Mann–Whitney tests for continuous parameters. Analyses were performed using Stata, version 15.1, College Station, TX, USA.

### 2.3. Laboratory Tests

We retrospectively performed tests on samples prospectively collected.

For the SARS-CoV-2 specific humoral response, SARS-CoV-2 IgG, IgM, and IgA levels were measured by ELISA according to manufacturer’s instructions (DIESSE Diagnostica Senese S.p.a., Monteriggioni, Italy) at three timepoints: 5–10 days (T1), 12–17 days (T2), and 90–120 days (T3) from the nasopharyngeal swab (NPS) positivity, respectively. The ratio between the optical density of the sample and that of the cutoff reagent (index) was calculated. The samples were scored as positive (index > 1.1), doubtful (index between 1.1 and 0.9), or negative (index < 0.9).

The evaluation of the neutralizing antibodies (nAbs) was performed using a SARS-CoV-2 microneutralization assay (MNA) based on live viruses (lineage B.1 clade G, GISAID accession number: EPI_ISL_568579), as previously described [21]. SARS-CoV-2 nAb titers are expressed as the reciprocal of serum dilution achieving 90% of neutralizing activity. Serum from the National Institute for Biological Standards and Control (NIBSC, UK) with known neutralization titer was used as the reference in MNA (Research reagent for anti-SARS-CoV-2 Ab NIBSC code 20/136).

For the inflammatory profile, plasma samples were obtained after speed centrifugation for 10 min at 2000 rpm and immediately stored at −80 °C. IL-1β, IL-6, IL-8, and TNF-α were measured in plasma samples using an automated ELISA assay (ELLA microfluidic analyzer, Protein Simple). In group 1, the SARS-CoV-2 T-cell response was analyzed by ELISpot assay at three different timepoints after molecular test: 5–10 days from the NPS positivity (T1), 12–17 days from positivity (T2), and 90–120 days from positivity (T3). Briefly, peripheral blood mononuclear cells (PBMCs) were isolated from whole blood using the Ficoll procedure, resuspended in culture medium (RPMI 1640, 10% FCS, 2 mM l-glutamine, 50 IU/mL penicillin, and 50 µg/mL streptomycin), counted, and stored at −80 °C. Cryopreserved PBMCs were rapidly thawed, washed with culture medium, assessed for vitality by Trypan Blue exclusion, and plated at 3 × 10^5^ cells/well in ELISpot plates (AID GmbH, Strabberg, Germany). PBMCs were then stimulated with 1 µg/mL of SARS-CoV-2 specific peptide pools of overlapping spike and nucleocapsid proteins (Peptides and Elephants, Hennigsdorf, Germany) and PHA, included in the ELISpot kit, for 24 h with 5% CO_2_. At the end of incubation, the ELISpot assay was developed according to the manufacturer’s instructions. Positive results are expressed as at least 20 spot-forming cells (SFCs)/10^6^ PBMCs in stimulating cultures after subtracting the spontaneous background.

## 3. Results

### 3.1. Characteristics of the Study Population

Table 1 shows the demographic characteristics at COVID-19 diagnosis or hospital admission of advanced HIV-positive patients with COVID-19 (group 1); comparisons of this group with advanced HIV-positive patients without COVID-19 (group 2) and with patients with COVID-19 without HIV infection (group 3) are also reported.

The median age of advanced HIV-positive patients coinfected with COVID-19 was 57 years old (IQR, 53–62), 76% male; the HIV transmission occurred mainly by unprotected sex, and the median HIV duration of infection was 18 years (15–29) with a nadir of CD4 of 57/mmc (23–100) and a current CD4 count of 348 (186–565). Furthermore, 83% of them had at least one comorbidity, mainly represented by central nervous system diseases (23/30; specifically, nine AIDS/dementia complex, two HIV encephalopathy, five progressive multifocal leukoencephalopathy, three cerebral ischemic sequelae, one hepatic encephalopathy, one hydrocephalus, one cerebral toxoplasmosis-related hemiplegia, and one demyelinating polyneuropathy). Additionally, 30% (9/30) of patients were on a tenofovir/disoproxil/fumarate (TDF)-based triple antiretroviral regimen, and 70% (21/30) were taking a boosted protease inhibitor (b/PI) as an anchor drug. The severity of COVID-19, according to NIH criteria [20], was asymptomatic in 70% and mild/moderate in 30%. The overall mortality rate with COVID-19 was 10% (3/30). The median time to viral clearance was 18 days (16–25).

#### SARS-CoV-2 Specific Immune Response

Focusing on HIV-infected patients with COVID-19, we observed that 90% of patients showed positive antibodies titers after a median of 26 days from the SARS-CoV-2 diagnosis. IgA, IgM, and IgG specific anti-SARS-CoV-2 kinetics evaluated at three timepoints (T1, T2, and T3; Figure 1) after COVID-19 diagnosis showed a different trend across the three antibody classes. IgM antibodies were the first to appear, showing very short kinetics and low titers (Figure 1A). IgA and IgG antibodies showed slightly longer kinetics of appearance and reached significantly higher titers at T2 than T1 (Figure 1B,C). Both IgM and IgA antibodies at T3 decayed until they almost disappeared, while IgG antibodies persisted stable until T3. The neutralizing activity increased significantly at T2 and then significantly dropped at T3. The peak of neutralizing titers at T2 was probably due to the protective activity of both IgA and IgG SARS-CoV-2-specific antibodies.

Specifically, early after infection (T1), 13 patients out of 25 (52.0%) showed positive IgM antibodies (Figure 1A), whereas only six and five patients out of 25 showed positive IgG and IgA antibodies, respectively (Figure 1B,C); only two patients out of 25 showed neutralizing antibodies at T1 (Figure 1D). At T2, both IgA and IgG antibodies peaked; then, at T3, IgA antibodies rapidly decreased, whereas IgG antibodies remained stable. All but three patients showed neutralizing antibodies at T2.

We did not find any correlation between humoral response in terms of IgG production and CD4 count at T1, T2, and T3. The kinetics of the SARS-CoV-2 specific T cells was also evaluated at three timepoints after NPS positivity (T1, T2, and T3; Figure 2A–C). Early after infection (T1), 10 patients out of 22 (45.5%) showed a positive response that significantly increased at T2 and then slightly decreased (median T1: 17.5 SFC/106 (7.0–120) vs. T2: 154 SFC/106 (45.2–272.3) vs. T3: 93.33 SFC/106 (13.33–144.2), *p* = 0.0045). All patients developed a positive specific T-cell response over time, but the kinetics was variable. Specifically, in the majority of patients (63.6%) the peak T-cell response occurred at T2, while it was delayed at T3 in the remaining patients (Figure 2B). Of note, a positive correlation between the strength of the T-cell response and the CD4 T-cell frequency was observed, highlighting the main role of CD4 T cells in the SARS-CoV-2 specific T-cell response (*p* = 0.0017; *R* = 0.5272) (Figure 2C).

### 3.2. Comparisons of the Characteristics of the Three Groups Included in the Analysis

For the purpose of this analysis, *n* = 30 advanced HIV patients with COVID-19 (group 1), *n* = 52 advanced HIV patients with no COVID-19 (group 2), and *n* = 58 COVID-19 patients without HIV infection (group 3) were included. By comparing groups 1 and 2, we observed a median age of 44 years old in group 1, lower than in group 2, while the HIV transmission occurred mainly by unprotected sex in both groups. Moreover, AIDS-defining diseases were found similarly in both groups 1 and 2 (67% vs. 60%), whereas the duration of HIV infection was 2 years shorter in group 1 than in group 2, and the nadir of CD4 count was higher in group 2 than in group 1. By comparing the two COVID-19 groups, i.e., group 1 with HIV coinfection and group 3 without HIV infection, we noted no significant differences in terms of age, number of comorbidities, SpO_2_ at COVID-19 diagnosis, and death. The differences we found pertained to the proportion of patients with symptoms and pneumonia, resulting higher in group 3 than in group 1, but with no differences in terms of severity (all COVID-19 cases without HIV infection were mild/moderate) and in terms of median time to viral clearance.

### 3.3. Inflammatory Biomarkers

The levels of inflammatory markers were quantified in all three groups (Figure 3). With the exception of IL-1β and TNF-α, HIV/COVID-19 patients showed similar levels of IL-6 and IL-8 than those observed in HIV patients without COVID-19. Of note, these cytokines (IL-6 and IL-8) were lower in HIV/COVID-19-coinfected individuals than in COVID-19 patients, suggesting a dampened inflammatory response to SARS-CoV-2 infection in immunocompromised subjects. In contrast, no significant differences in the levels of IL-1β and TNFα were observed between these two groups.

We did not find any significant correlation between inflammatory cytokines and CD4 count (for IL-1β, B −2.69 (−11.3; 5.95), *p* = 0.529; for IL-6, B −0.04 (−0.37; 0.29), *p* = 0.800; for TNF-α, B −0.47 (−1.17;0.22), *p* = 0.177), with the exception of a signal coming from IL-8 (B −1.48 (−2.98; 0.03), *p* = 0.054).

## 4. Discussion

Our experience coming from the cohort of PLWH and COVID-19 in a rehabilitation facility indicates that these patients, with a long course of HIV infection and medical comorbidities, remain at risk of COVID-19 even though they did not show a more severe clinical course than patients without HIV coinfection.

As more data are emerging, we have learned that advanced HIV disease stage and unsuppressed viral load (VL) are significantly associated with a high hospitalization risk [15]; moreover, smaller sized studies from the UK and France asserted that there might be a consistently higher morbidity and mortality risk from COVID-19 among virologically suppressed black PLWH [22,23]. Interestingly, a pronounced immunodeficiency (current CD4 count < 350/µL) was found to be associated with an increased risk for severe COVID-19 (aOR 2.85, 95% CI: 1.26–6.44, *p* = 0.01) and low nadir CD4 counts associated with an increased mortality [24]. In addition to epidemiological studies, data on clinical and immunologic outcomes of PLWH are necessary to understand the manifestations of this novel coinfection. It is also known that perturbations in T-cell subsets might affect the cellular immune response to SARS-CoV-2 [25,26], and PLWH may be particularly vulnerable to such effects given residual immune dysregulation even in the presence of suppressive ART [27]. The current median value of the CD4 count of our cohort was below the aforementioned threshold of 350 cells/µL, and the nadir of CD4 was very low; nevertheless, they seemed to be less affected by severe COVID-19 manifestations than non-HIV-infected patients with a broadly similar mortality rate. Additionally, 90% of patients showed positive antibody titers after a median of 26 days from the SARS-CoV-2 infection diagnosis, and neutralizing activity was already observed 2–3 weeks after the NPS positivity with a peak probably due to the protective activity of both IgA and IgG antibodies. Similarly, all patients developed a positive specific T-cell response over time even if with variable kinetics, with the majority of patients (63.6%) reaching the peak T-cell response about 2–3 weeks after the positivity; interestingly, this result correlated with the CD4 counts.

These data confirm what is currently being observed with vaccine responses [28]; virologically suppressed HIV-positive patients, even if advanced and with comorbidities, are able to develop a humoral and neutralizing response, a prerequisite for a satisfactory vaccine response. Groups 1 and 3 were not comparable in terms of severity or treatment (corticosteroids) and, therefore, inflammatory biomarkers.

Despite the criterion of comparison between group 1 and group 3 based on the NIH Disease Severity Clinical Progression Scale for COVID-19 definition, they are actually not fully comparable. First, HIV patients cannot be compared with the HIV-negative population of the same age because of the phenomena of premature aging that characterizes the condition of chronic HIV infection. This is why, even now, there is still no univocity on the higher risk of disease severity in the HIV population compared to the HIV-negative population. In addition, the prevalence of confirmed pneumonias in group 1 was significantly lower than that in group 3; however, given the characteristics of the host center for group 1 patients, we cannot rule out a higher prevalence of pneumonias.

Inflammation pattern was found to be related to disease severity in COVID-19 [29], and a recent study conducted in New York suggested that PLWH hospitalized for COVID-19 remain able to mount an inflammatory reaction in response to SARS-CoV-2 coinfection [30], similar to that observed in other populations [31]. In that study [30], levels of inflammatory biomarkers, C-reactive protein (CRP), fibrinogen, D-dimer, interleukin 6 (IL-6), interleukin 8 (IL-8), and tumor necrosis factor α (TNFα) were commonly elevated, with the exception of IL-1β.

Although an interesting signal came from the correlation between inflammatory cytokines and CD4 count, we were unable to draw definitive conclusions, and further studies on larger sample sizes are needed to confirm this finding. In our study, a lower inflammatory response to COVID-19 was observed in HIV-positive patients than in HIV-negative subjects. In particular, IL-6 and IL-8 were expressed significantly less in HIV/COVID-19 patients than in COVID-19 patients without HIV, suggesting a possible impact of immunosuppression on dampening the inflammatory response.

Comparing the levels of inflammatory cytokines between the group with HIV infection and the group with HIV/COVID-19 coinfection, we noted that IL-6 and IL-8 did not differ significantly. On the contrary, we saw a difference in terms of TNF-α levels (higher in HIV/COVID-19 than in HIV group) and IL-1β (higher in the HIV group than in the group of coinfected participants). The relationship between inflammation markers with morbidity/mortality from non-AIDS events in treated HIV infection is known, and the chronic immune activation characterizing these patients may be even more associated with adverse clinical outcomes than in the general population [32]. For these reasons, we considered it relevant to compare the inflammation levels of chronically infected HIV patients with COVID-19 and those without COVID-19, as well as COVID-19 cases without HIV infection. Our findings demonstrate that HIV chronic infection on ART, in an advanced stage, does not seem to negatively impact the COVID-19-related inflammatory state, which nevertheless remains attenuated overall if compared to COVID-19 without HIV coinfection.

A research team from France recently reported that, in 30 patients with HIV/SARS-CoV-2 coinfection, HIV infection was probably not an independent risk factor for COVID-19 [33]; another remarkable study from Russia strongly supported this point of view, in which HIV infection with suppressed HIV RNA does not represent a serious comorbidity for COVID-19, whereas SARS-CoV-2 infection does not influence the immunity in such patients more than the immunity of the general population [34].

Additionally, the specific immune response and antibody production in patients with HIV and SARS-CoV-2 coinfection does not seem to be damaged, with IgG and neutralizing antibody titers persisting 3–4 months after positivity and reaching an early peak within 2–3 weeks after NPS, in addition to a SARS-CoV-2 specific T cell response.

This study had some limitations. Firstly, this work was the result of a collaboration between the National Institute for Infectious Diseases L. Spallanzani, IRCCS, in Rome and a drug rehabilitation center that also houses a hospital section with advanced HIV patients who are mostly poorly self-sufficient due to AIDS-related neurological conditions. During the second wave of the pandemic, this facility was used as a low-intensity care COVID hospital. Since the latter is not a full-fledged hospital, it is possible that patients were evaluated clinically but without always performing, at SARS-CoV-2 positivity, a diagnostic imaging test that could diagnose the presence of pneumonia. Our institute, on the other hand, is a reference COVID-19 hospital that performs a chest X-ray or chest CT scan on every admission. Therefore, we decided to compare the two groups on the basis of clinical characteristics, i.e., having a mild/moderate COVID-19 according to the NIH Clinical Progression Scale for COVID-19 definition (available online at https://www.covid19treatmentguidelines.nih.gov/overview/clinical-spectrum/ (accessed on 1 June 2021)). For this reason, we cannot rule out that the COVID-19/HIV group did not have pneumonia. Secondly, we were not able to provide the specificity, intensity, and clinical efficacy of these responses. In spite of the many limitations of this work, our findings give additional information on the clinical and immunological outcomes of a cohort of advanced HIV-infected individuals who are not spared from SARS-CoV-2 contagion by living in a community where distancing measures are difficult to implement. Moreover, these data suggest that PLWH might successfully benefit from anti-SARS-CoV-2 vaccination, which should be recommended for all PLWH, particularly for those with a low CD4 count. Further data are mandatory to evaluate the durability of these immunity and its ability to be implemented after viral exposure and/or anti-SARS-CoV-2 vaccination.

## Figures and Tables

**Figure 1 viruses-14-01575-f001:**
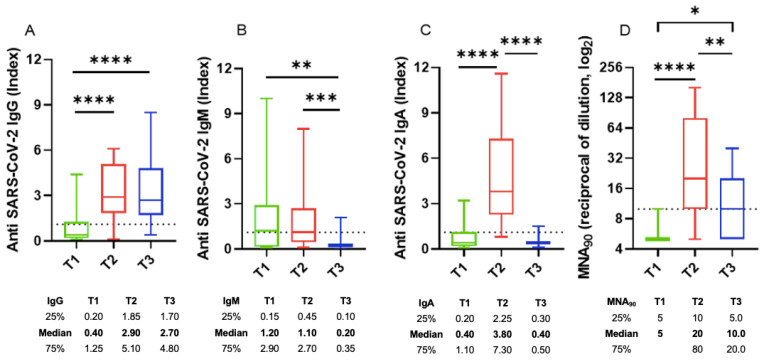
SARS-CoV-2 specific antibody kinetics in group 1 patients (*n* = 25): specific (**A**) IgG, (**B**) IgM, and (**C**) IgA titers at T1, T2, and T3; the dotted line indicates the limit of Ig quantification. (**D**) Neutralizing antibody titers at T1, T2, and T3; the dotted line indicates the limit of neutralizing antibody quantification. (**A**) **** means *p* < 0.0001, (**B**) ** means *p* = 0.0061 and *** means *p* = 0.0002, (**C**) **** means *p* < 0.0001, (**D**) * means *p* = 0.0201, ** means *p* = 0.0092, and **** means *p* < 0.0001 according to Kruskal–Wallis test.

**Figure 2 viruses-14-01575-f002:**
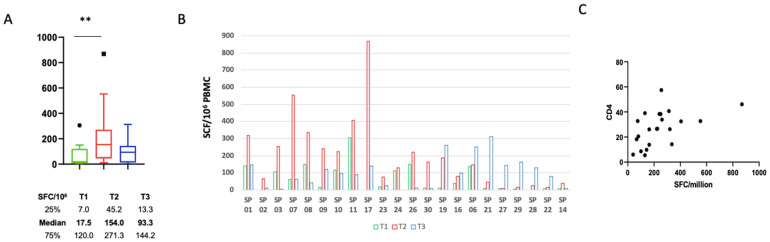
Specific SARS-CoV-2 specific T-cell kinetics: (**A**) SARS-CoV-2 specific T-cell response at T1, T2, and T3, expressed as spot-forming cells (SFCs)/10^6^ PBMCs. ** means *p* < 0.01 according to Kruskal–Wallis test. (**B**) SARS-CoV-2 specific T-cell response in each HIV-coinfected subject at T1, T2, and T3. (**C**) Correlation between the absolute number of circulating CD4 T cells and SCFs/10^6^ PBMCs (Spearman test: *p* = 0.0017; *R* = 0.5272).

**Figure 3 viruses-14-01575-f003:**
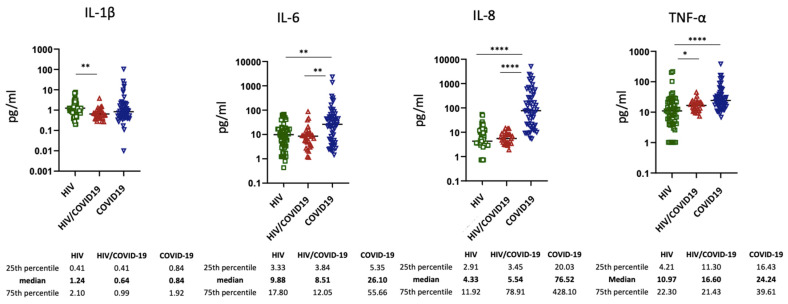
Plasmatic IL-1, IL-6, IL-8, and TNF-a in HIV (green circles), HIV/COVID-19 (red circles), and COVID-19 (blue circles) subjects according to ELISA assays. Each dot represents one single participant, and the horizontal bar identifies the median values for each cytokine. The details of the median and IQRs are also shown; * means *p* = 0.05, ** means *p* = 0.002; **** means *p* < 0.0001.

**Table 1 viruses-14-01575-t001:** General characteristics of study population.

General Characteristics	HIV Positive COVID-19	HIV Positive No COVID-19	*p*-Value ^#^	COVID-19 No HIV	*p*-Value *
	Group 1	Group 2		Group 3	
	*n* = 30	*n* = 52		*n* = 58	
**Gender, *n* (%)**					
M	23 (76.7)	44 (84.6)	0.332	42 (72.4)	0.667
**Age, years, median (IQR)**	57 (53–62)	49 (43–53)	<0.001	58 (42–71)	0.817
**Mode of transmission of HIV, *n* (%)**					
Unprotected sex	18 (60.0)	24 (46.2)	0.069		
IVDUs	12 (40.0)	20 (38.5)			
Unknown	-	8 (15.4)			
**Years of HIV infection, median (IQR)**	18 (15–29)	16 (9–20)	0.026		
**Nadir of CD4 count, cell/mmc, median (IQR)**	57 (23–100)	92 (52–149)	0.046		
**CD4 count at test, cell/mmc, median (IQR)**	348 (186–565)	316 (180–525)	0.597		
**CD8 count at test, cell/mmc, median (IQR)**	756 (563–1124)	600 (384–917)	0.064		
**Antiretroviral treatment, *n*(%)**	30 (100)	52(100)			
**At least 1 comorbidity, *n* (%)**	25 (83.3)			38 (65.5)	0.079
**COVID-19 symptoms, *n* (%)**	9 (30.0)			54 (93.1)	<0.001
**COVID-19 pneumonia, *n* (%)**	6 (30)			50 (86.2)	<0.001
**SpO_2_ at COVID-19 diagnosis/hospital admission, median (IQR)**	96 (96–96)			96 (95–98)	0.542
**Death with COVID-19, *n* (%)**	3 (10.0)			9 (15.5)	0.475
**Time to viral clearance, days, median (IQR)**	18 (16–25)			17 (10–25)	0.137
**Steroid treatment, *n* (%)**	6 (20.0)			30 (51.7)	0.004
**Remdesivir, *n* (%)**	4 (13.3)			5 (8.6)	0.489

^#^ Comparison of group 2 vs. group 1; * comparison of group 3 vs. group 1.

## Data Availability

The dataset will be made available upon reasonable request.

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
