# Peer review of "SARS-CoV-2 Specific Immune Response and Inflammatory Profile in Advanced HIV-Infected Persons during a COVID-19 Outbreak"

_viruses, 2022, doi:10.3390/v14071575_

Round 1

Reviewer 1 Report

Vergori A and colleagues explored retrospectively immune responses in HIV+ patients (treated and stabilized) during COVID-19. Most of these patients had mild forms of the disease (COVID-19 without pneumonia, only 6 patients (30%) with pneumonia) while the majority of COVID-19 controls HIV- suffered from pneumonia (86.2%). The main results of this work are:

- Most (90%) HIV+ patients produced anti-SARS-CoV-2 antibodies (in particular IgG and neutralizing Ab) in the first month following infection (humoral response)

- Almost all HIV+ patients (monitored and treated) showed a cellular response (T lymphocytes and in particular TCD4) against SARS-CoV-2 in the 1st month.

This work is correctly made but not very original and above all does not allow any conclusion as to the specific anti-SARS-CoV-2 immunity in HIV+ patients. Major revisions may be needed.

1) The majority of controls should have a non-severe form (no pneumonia) of COVID-19 to make the comparison meaningful.

The discussion must imperatively underline this major limit to the interpretation of the study.

2) While it is interesting to observe a cellular and humoral anti-SARS-CoV-2 response in the majority of HIV+ patients, the specificity, intensity and efficacy of these responses is impossible to specify

Immunological explorations over time (at 3 months, 6 months, etc.) would be interesting to verify the persistence of the T response and neutralizing antibodies.

Antibody production and the T response should be compared to the response in controls that are not infected with HIV (groupe 3). Warning: as in 1), the controls must be COVID+ patients who present a comparable severity of the disease! (Mild). Which is not the case at the moment.

Other details to be provided are as follows:

1) What is the legal framework for blood samples? Permissions are specified for group 3 but not clear for group 1.

2) Regarding the results:

- Discuss a suppression of group 2 (which for me has no practical interest)

- Replace the SpO2 (which depends in particular on the possible oxygen support) by the need (or not) for an oxygen support specifying this support (conventional oxygen, high flow oxygen therapy, invasive mechanical ventilation)

- Why do 83% of patients in group 1 have a comorbidity and especially a neurological one ? Does your center specialize in this care? These items need to be specified. The existence of neurological comorbidities (in particular cognitive disorders) makes the precision of the legal framework of the study even more essential.

- The presentation of the results should change order: 1 = characteristics of the groups, 2 = inflammatory biomarkers (innate immunity), cellular immunity and then, humoral immunity and antibodies.

- We would like to know the specific prognosis (mortality, length of hospital stay) of HIV+COVID+ patients who do not present a humoral and/or cellular response in the 1st month: is there a correlation with the prognosis? and a correlation between the 2 types of immune responses ?

Author Response

Vergori A and colleagues explored retrospectively immune responses in HIV+ patients (treated and stabilized) during COVID-19. Most of these patients had mild forms of the disease (COVID-19 without pneumonia, only 6 patients (30%) with pneumonia) while the majority of COVID-19 controls HIV- suffered from pneumonia (86.2%). The main results of this work are:

- Most (90%) HIV+ patients produced anti-SARS-CoV-2 antibodies (in particular IgG and neutralizing Ab) in the first month following infection (humoral response)

- Almost all HIV+ patients (monitored and treated) showed a cellular response (T lymphocytes and in particular TCD4) against SARS-CoV-2 in the 1st month.

This work is correctly made but not very original and above all does not allow any conclusion as to the specific anti-SARS-CoV-2 immunity in HIV+ patients. Major revisions may be needed.

1) The majority of controls should have a non-severe form (no pneumonia) of COVID-19 to make the comparison meaningful.

Thank you for the comment.

The work is the result of a collaboration between the National Institute for Infectious Diseases in Rome and a drug rehabilitation centre that also houses a hospital section with advanced HIV patients mostly poorly self-sufficient due to AIDS related neurological conditions but not only. During the second wave pandemic this facility has been used as a low-intensity care COVID hospital. Since the latter is not a full-fledged hospital, it is possible that patients were evaluated clinically but without always performing at SARS-CoV-2 positivity a diagnostic imaging test that could diagnose the presence of pneumonia. Ours, on the other hand, is a referee COVID  Hospital that performs Chest X-ray or Chest CT scan on every admission.

Therefore, we decided to compare the two groups on the basis of clinical characteristics, namely having a mild/moderate COVID19 based on the NIH Clinical Progression Scale for COVID-19 definition (available at https://www.covid19treatmentguidelines.nih.gov/overview/clinical-spectrum/).

At this point, performing an analysis that excludes the COVID HIV- group's pneumonia cases is not feasible in our opinion, because it cannot be excluded that the COVID HIV+ group does not have pneumonia.

We have now specified these characteristics as the main limitation in the manuscript.

2) While it is interesting to observe a cellular and humoral anti-SARS-CoV-2 response in the majority of HIV+ patients, the specificity, intensity and efficacy of these responses is impossible to specify

We agree,  indeed, to date, no conclusive evidence about clinical outcomes in  COVID-19 and HIV infection is available. In fact, many small cohort studies failed to demonstrate higher risk rate of COVID-19 or a more aggressive course of disease in people living with HIV (PLWH) in comparison to those without HIV (non-PLWH), although larger and more recent studies reported a higher risk of poor outcomes of COVID-19 in PLWH, in particular a higher mortality rate, higher rate of hospitalization due to COVID-19  or higher rate of  in-hospital intubation. But nevertheless, limited information about difference in levels of inflammatory and immune markers among people hospitalized with COVID-19 by HIV serostatus is available. Moreover, also recent studies on immunogenicity of SARS-CoV-2 vaccine are not able to provide data on specificity, intensity and efficacy data in terms of breakthrough infections due to the small number of infections in this category of patients.

Immunological explorations over time (at 3 months, 6 months, etc.) would be interesting to verify the persistence of the T response and neutralizing antibodies.

Thank you for the suggestion,  but since this is a work on immunity from natural SARS-CoV-2 infection, immunological exploration over time it’s not feasible anymore given the now complete vaccination of the patients in study.

Antibody production and the T response should be compared to the response in controls that are not infected with HIV (groupe 3). Warning: as in 1), the controls must be COVID+ patients who present a comparable severity of the disease! (Mild). Which is not the case at the moment.

This comparison is absolutely interesting, unfortunately, however, we could not retrospectively test patients in group 3 due to lack of available samples. In fact, the work aims, not so much to compare as to describe the HIV/COVID outbreak, and collaterally to compare only the inflammatory aspect.

Other details to be provided are as follows:

1) What is the legal framework for blood samples? Permissions are specified for group 3 but not clear for group 1.

San Patrignano Centre is a rehabilitation facility in which, at admission, people sign a release on the use of their data and samples, in anonymous and aggregate form, for research or statistical processing purposes. For this specific study, ethical review and approval were not required for the study on human participants in accordance with the local legislation and institutional requirements. The patients/participants or their care givers, as appropriate, provided additional written informed consent to participate in this study.

2) Regarding the results:

- Discuss a suppression of group 2 (which for me has no practical interest)

The role of group 2 is to provide a comparison of a chronic infection (HIV) known to induce a constant, chronic level of inflammation and immune-activation with an acute infection such as COVID19 and then compare these single infections with those who have both (group 1) instead. The discussion is in fact deficient in this argument, so we thank you for the suggestion and have taken care to include this part in the manuscript.

-Replace the SpO2 (which depends in particular on the possible oxygen support) by the need (or not) for an oxygen support specifying this support (conventional oxygen, high flow oxygen therapy, invasive mechanical ventilation)

SpO2 reported is that recorded at COVID19 diagnosis/hospital admission on room air, with the intent to include patients with mild/moderate infection according to the NIH severity scale as reported and specified above. This has been specified in the results section.

- Why do 83% of patients in group 1 have a comorbidity and especially a neurological one ? Does your center specialize in this care? These items need to be specified. The existence of neurological comorbidities (in particular cognitive disorders) makes the precision of the legal framework of the study even more essential.

San Patrignano centre is drug rehabilitation facility that also houses a hospital section with advanced HIV patients mostly poorly self-sufficient due to AIDS related neurological conditions but not only. During the first wave pandemic this facility has been used as a low-intensity care COVID hospital for the same patients. The legal framework has been specified above.

- The presentation of the results should change order: 1 = characteristics of the groups, 2 = inflammatory biomarkers (innate immunity), cellular immunity and then, humoral immunity and antibodies.

We respectfully disagree with the reviewer,  the main aim of the study was to describe the  HIV/COVID19 patients of the outbreak in the rehabilitation facility, secondary objectives were comparisons on inflammatory pattern.

- We would like to know the specific prognosis (mortality, length of hospital stay) of HIV+COVID+ patients who do not present a humoral and/or cellular response in the 1st month: is there a correlation with the prognosis? and a correlation between the 2 types of immune responses ?

Thank you for the interesting suggestion, we have now analysed the association of the humoral and cellular immunity with mortality and duration of viral shedding. As these patients are hospitalized in the rehabilitation facility, we do not have data on length of hospital stay, so as a surrogate of recovery, we used the duration of viral shedding.

For humoral immunity:

-the non-response in terms of absence of IgG at T1 are not associated with an higher risk of mortality (only N=2 deaths) [OR 0.42 (95%CI 0.02-7.59);p=0.558]; at T2 and T3 not calculable for absence of events.

-the non-response in terms of absence of IgG at T1 are not correlated with an longer duration of viral shedding at T1 [B 0.19 (95%CI -1.13;1.52);p=0.766]; at T2 we found a correlation with a shorter duration of the viral shedding [B -1.37 (95%CI -2.50;-0.23);p=0.020]; not correlated at T3, because all patients already negative for PCR SARS-CoV-2 on NPS.

For cellular immunity it was not possible to calculate the association with mortality due to absence of events on the basis of the  blood samples available and tested.

-the non response in terms of absence of at least 20 spots forming cells (SFC)/106 PBMCs

- the non response in terms of absence of at least 20 spots forming cells (SFC)/106 PBMCs at T1 are not associated with a different duration of the viral shedding at T1  [B -0.02 (95%CI -0.02-0.06);p=0.315], at T2 [B -0.002 (95%CI -0.02-0.01);p=0.817] and at T3 because all patients already negative for PCR SARS-CoV-2 on NPS.

Although interesting signals come from these correlations, we are unable to draw conclusions and to report these findings with any degree of certainty, given the small sample size and unadjusted nature of the analysis.

Reviewer 2 Report

It is a very interesting work, adding relevant information for a better managment of COVID 19 in HIV persons in terms of vaccine development and disease managment.

It would be interesting to see if there is any correlation between CD4 counts and antibody titers, or between CD4 counts and cytokine concentrations.

Methods: please indicate the positivity criteria used for ELISPOT assays. 

You could refer to the results described in the following manuscript: J Infect. 2022 May 27;S0163-4453(22)00315-2.  doi: 10.1016/j.jinf.2022.05.026. Online ahead of print.  SARS-CoV-2 humoral and cellular immune responses in COVID-19 convalescent individuals with HIV.

Figure 3: Typesetting error - HIV/COVI-19 in IL-6 graph. Please correct

Author Response

It is a very interesting work, adding relevant information for a better managment of COVID 19 in HIV persons in terms of vaccine development and disease managment.

It would be interesting to see if there is any correlation between CD4 counts and antibody titers, or between CD4 counts and cytokine concentrations.

Thank you for the interesting suggestion, we have now analysed the correlations as suggested.

 For CD4 count and antibody titres at T1: B 1.94 (-1.28;5.15); p=0.227; at T2: B 0.37 (-2.70; -3.43); p=0.808.

For CD4 count an cytokines:

IL1b B -2.69 (-11.3;5.95);p=0.529

IL6 -0.04 (-0.37;0.29);P=0.800

IL8 -1.48 (-2.98;0.03);P=0.054

TNFa -0.47 (-1.17;0.22);p=0.177

We have now incorporated these findings in results section.

Methods: please indicate the positivity criteria used for ELISPOT assays. 

Positive results are expressed as at least 20 spots forming cells (SFC)/106 PBMCs in stimulating cultures after subtracting spontaneous background. It has now been specified in the text.

You could refer to the results described in the following manuscript: J Infect. 2022 May 27;S0163-4453(22)00315-2.  doi: 10.1016/j.jinf.2022.05.026. Online ahead of print.  SARS-CoV-2 humoral and cellular immune responses in COVID-19 convalescent individuals with HIV.

Figure 3: Typesetting error - HIV/COVI-19 in IL-6 graph. Please correct

The error has been amended

Reviewer 3 Report

In this study, Vergori and colleagues analyzed the clinical parameters and inflammatory cytokine levels in plasma collected from age-controlled COVID19 patients who are either HIV naïve or living with HIV. In addition, with the emphasis on HIV/SARS CoV 2 co-infection, the dynamics of humoral and cellular responses targeting SARS CoV 2 were characterized at three times points. Overall, this study is well conceptualized and itself provides some incremental advances to our understanding of clinical characteristics and immunological events in COVID19 individuals living with HIV. However, the manuscript could be substantially improved with some more clarifications and language corrections.

Comments:

1. Please clarify the inconsistency between the context (line 1 04 110) and Table 1 It appears the COVID19 symptom/pneumonia rates are different between groups 1 and 3. With a significant difference in disease severity between these two groups, I am concerning biased subject recruitment, which would possibly explain the differential levels of IL 6 and IL 8 in figure 3.

2. In either the main text or the legend of figure 1, please indicate the exact number of samples tested for each antibody response.

3. For the T cell IFN gamma production assay, what is the positivity criterion? Moreover, please indicate data from what time points were used in Figure 2C? Also, I am curious about what contributes to the delayed peak of T cell response in certain patients as compared to the others. Are there any clinical differences between these two subgroups, i.e. CD4 counts, and viral load?

4. I do suggest an improvement of English for the present manuscript. Here are some examples:

a) Line 129, titres

b) Please make the exponent s superscript in lines 143, 148 etc.

c) Line 291, poin

d) Please unify the nomenclature s, for example, ELISpot and Elispot.

e) It would be better to reorganize the abstract according to the result section in the main text

f) And I do not think it is appropriate to conclude in a scientific journal by experience (Line 238)

g) Line 295, it could be either the SARS CoV 2 specific T cell-based immune response and antibody production or just the SARS CoV 2 specific immune response

Author Response

In this study, Vergori and colleagues analyzed the clinical parameters and inflammatory cytokine levels in plasma collected from age-controlled COVID19 patients who are either HIV naïve or living with HIV. In addition, with the emphasis on HIV/SARS CoV 2 co-infection, the dynamics of humoral and cellular responses targeting SARS CoV 2 were characterized at three times points. Overall, this study is well conceptualized and itself provides some incremental advances to our understanding of clinical characteristics and immunological events in COVID19 individuals living with HIV. However, the manuscript could be substantially improved with some more clarifications and language corrections.

Comments:

  1. Please clarify the inconsistency between the context (line 1 04 110) and Table 1 It appears the COVID19 symptom/pneumonia rates are different between groups 1 and 3. With a significant difference in disease severity between these two groups, I am concerning biased subject recruitment, which would possibly explain the differential levels of IL 6 and IL 8 in figure 3.

The work is the result of a collaboration between the National Institute for Infectious Diseases in Rome and a drug rehabilitation centre that also houses a hospital section with advanced HIV patients mostly poorly self-sufficient due to AIDS related neurological conditions but not only. During the second wave pandemic this facility has been used as a low-intensity care COVID hospital. Since the latter is not a full-fledged hospital, it is possible that patients were evaluated clinically but without always performing, at SARS-CoV-2 positivity, a diagnostic imaging test that could diagnose the presence of pneumonia. Ours, on the other hand, is a referee COVID  Hospital that performs Chest X-ray or Chest CT scan on every admission. Therefore, we decided to compare the two groups on the basis of clinical characteristics, namely having a mild/moderate COVID19 based on the NIH Clinical Progression Scale for COVID-19 definition (available at https://www.covid19treatmentguidelines.nih.gov/overview/clinical-spectrum/).

  1. In either the main text or the legend of figure 1, please indicate the exact number of samples tested for each antibody response.

As suggested this information was reported both in the main text (Results/paragraph: Humoral and cell-mediated immune response to SARS-CoV-2, now renamed SARS-CoV-2 specific immune response according to your suggestio ) and in the legend of figure 1.

  1. For the T cell IFN gamma production assay, what is the positivity criterion? Moreover, please indicate data from what time points were used in Figure 2C? Also, I am curious about what contributes to the delayed peak of T cell response in certain patients as compared to the others. Are there any clinical differences between these two subgroups, i.e. CD4 counts, and viral load?

Figure 2C shows the correlation between the number of CD4 T cells and the maximum level of antigen-specific T cell response. In some patients the peak was early (T2), in others it was later (T3).

We agree with the reviewer that the different kinetic of T cell response is an interesting feature. Unfortunately, the low number of patients did not allow to highlight any specific differences between the “early” and “late” responders. We can speculate that different degree of both numerical and functional immunorecovery might impact the rapidity of the immune response.

  1. I do suggest an improvement of English for the present manuscript. Here are some examples:
  2. a) Line 129, titres
  3. b) Please make the exponent s superscript in lines 143, 148 etc.
  4. c) Line 291, poin
  5. d) Please unify the nomenclature s, for example, ELISpot and Elispot.
  6. e) It would be better to reorganize the abstract according to the result section in the main text
  7. f) And I do not think it is appropriate to conclude in a scientific journal by experience (Line 238)
  8. g) Line 295, it could be either the SARS CoV 2 specific T cell-based immune response and antibody production or just the SARS CoV 2 specific immune response

As suggested, all the changes have been made.

Round 2

Reviewer 1 Report

Groups 1 and 3 are not comparable in terms of severity, treatment (corticosteroids) and therefore inflammatory biomarkers. This should be mentioned clearly in the discussion (lines 293-297).

Author Response

Thank you again for your suggestion. We have now added a specific statement on this issue in discussion section, specifying why the two groups cannot be compared.